# Causal Association between Iritis or Uveitis and Glaucoma: A Two-Sample Mendelian Randomisation Study

**DOI:** 10.3390/genes14030642

**Published:** 2023-03-03

**Authors:** Je Hyun Seo, Young Lee

**Affiliations:** 1Veterans Medical Research Institute, Veterans Health Service Medical Center, Seoul 05368, Republic of Korea; 2Department of Applied Statistics, Chung-Ang University, Seoul 06974, Republic of Korea

**Keywords:** glaucoma, Mendelian randomisation, iritis, uveitis, single-nucleotide polymorphisms

## Abstract

Recent studies have suggested an association between iritis or uveitis and glaucoma. This study investigated the causal relationship between glaucoma and iritis and uveitis as exposures in a multi-ethnic population. Single-nucleotide polymorphisms associated with exposures to iritis and uveitis from the genome-wide association study (GWAS) data of Biobank Japan (BBJ) and the meta-analysis data from BBJ and UK Biobank (UKB) were used as instrumental variables (IVs). The GWAS dataset for glaucoma was extracted from the meta-analysis data (*n* = 240,302) of Genetic Epidemiology Research in Adult Health and Aging and UKB. The casual estimates were assessed with a two-sample Mendelian randomisation (MR) test using the inverse-variance-weighted (IVW) method, weighted median method, MR–Egger method, and MR-Pleiotropy Residual Sum and Outlier test. The IVW method revealed a significant causal association between iritis and glaucoma using IVs (*p* < 5.0 ×$\;10^{-8}$) from the East Asian population (*n* = 2) (odds ratio [OR] = 1.01, *p* = 0.017), a significant association between iritis exposures (*p* < 5.0 × $10^{-8}$) in the multi-ethnic population (*n* = 11) (OR = 1.04, *p* = 0.001), and a significant causal association between uveitis exposures (*n* = 10 with *p* < 5.0 ×$\;10^{-8}$) and glaucoma in the multi-ethnic population (OR = 1.04, *p* = 0.001). Iritis and uveitis had causal effects on glaucoma risk based on IVs from the multi-ethnic population. These findings imply that the current classifications of uveitic glaucoma and open-angle glaucoma overlap, indicating the need for further investigating these complex relationships.

## 1. Introduction

Uveitis is an inflammatory condition that affects the middle layer of the eye wall [1,2]. Anterior uveitis, also known as iritis, is the most prevalent form of uveitis. A consequence of uveitis is glaucoma, a leading cause of irreversible vision loss characterized by the degeneration of the retinal ganglion cells and their axons [3]. The classification of glaucoma is based on anatomical and known-causal factors, with open-angle glaucoma (OAG) being the most common form according to the anatomical mechanism [4]. Uveitic glaucoma (UG) is a complex range of disorders characterized by the coexistence of iritis and glaucomatous optic neuropathy, which encompasses several diverse clinical entities with varying prognoses [5,6,7]. Nevertheless, not all patients with uveitis develop glaucoma, which can be caused by uveitis itself or occur as a side effect of treating inflammation. Previous studies have reported that the prevalence of glaucoma in uveitis ranges from 11–20% after 5 years [2,8,9,10]. 

Numerous studies have shown that multiple factors are involved in the pathophysiology of glaucoma; however, its pathogenesis is not fully understood [2,11]. Alterations in the intraocular pressure (IOP) are correlated with the development of glaucoma [11]. IOP elevation in UG occurs due to the obstruction of aqueous outflow due to changes in the trabecular meshwork tissue architecture or open-angle mechanisms [12] and angle closure mechanisms, such as pupillary block. Despite the complexity of the categorization and mechanism of UG, it is possible to discern whether the mechanism is open- or closed-angle based on the structure. Early stages of UG may present with open-angle and shared features of OAG. As uveitis and glaucoma are complex diseases with genetic predisposition and predisposing environmental factors [5,6,7,13,14,15,16,17,18,19], further research using genetic data is required to gain additional information regarding the association between uveitis and OAG.

Recent investigations on iritis and uveitis have indicated that uveitis is characterised by a strong relationship with the human leucocyte antigens (*HLA*) gene. Moreover, acute anterior uveitis (AAU), Behcet’s disease (BD), and Vogt–Koyanagi–Harada disease were found to be highly correlated with *HLA-B27*, *HLA-B51*, and *HLA-DR4/DQA1,* respectively [5,13]. The most recent genetic research data and information regarding single-nucleotide polymorphisms (SNPs) for glaucoma in a multi-ethnic population were obtained from a meta-analysis of the Genetic Epidemiology Research in Adult Health and Aging (GERA) cohort and UK Biobank (UKB) [20]. The disease pathophysiology and epidemiology can be studied by applying bioinformatics and statistical techniques to these data, and our study group recently evaluated a genetic risk score model based on the genome-wide association study results of OAG [21] and various conditions [22,23,24]. The causal effect of iritis and uveitis on glaucoma can be analysed by calculating the ratio of iritis-related SNPs and uveitis-related SNPs in the genetic data of patients with glaucoma. Mendelian randomisation (MR) analysis is a methodology used in genetic epidemiology that employs genetic variants linked with putative risk factors (exposures) as instrumental variables (IVs) to analyse the causal impact of these factors on clinical outcomes [25,26]. These techniques have been used in numerous studies that investigated the impact of risk factors, such as type 2 diabetes and the effect of coffee on glaucoma onset [27,28]. Thus, it may be possible to confirm whether uveitis and iritis are causal factors of glaucoma using MR analysis based on the calculated risk ratio or odds ratio (OR). In addition, it is anticipated that the selection of exposure factors from a large cohort of Biobank Japan (BBJ) and UKB data will lead to more significant results, which would provide a basis for recognizing iritis and uveitis as causal factors of glaucoma based on genetic data and aid in understanding UG pathology, as the pre-existing classification of glaucoma is a combination of anatomy and aetiology. For this purpose, we performed a two-sample MR test using summary data from BBJ for an East Asian population, a meta-analysis of BBJ and UKB for a multi-ethnic population [29], and glaucoma genetic data from a meta-analysis of GERA and UKB [20]. 

## 2. Materials and Methods

### 2.1. Study Design Overview

The institutional review board of the Veterans Health Service Medical Center approved this study protocol (IRB No. 2022-12-007) and waived the need for informed consent due to its retrospective nature. The study was conducted in compliance with the Declaration of Helsinki. 

### 2.2. Data Source 

A schematic plot of the design of this analytical study is shown in Figure 1. To explore the causal effect of exposure to iritis and uveitis on the risk of glaucoma, we selected datasets for SNP-related iritis and SNP-related uveitis as the exposures (Table 1) from the summary statistics of the genome-wide association study (GWAS) for (1) the East Asian population from BBJ (*n* = 175,653 for iritis, *n* = 174,725 for uveitis), and (2) the meta-analysis of the multi-ethnic population (*n* = 656,395 for iritis, *n* = 655,467 for uveitis) from BBJ and UKB [29]. The OAG summary statistics of the GWAS data were adopted from the meta-analysis (*n* = 240,302; 12,315 cases and 227,987 control) of GERA and UKB [30]. The OAG dataset was used in this study as most of the genetic data related to glaucoma have been investigated for OAG. Since most early stage UG cases are OAG, it was assumed that UG and OAG are classified on the basis of aetiology and angle shape, respectively [2]. Moreover, since this study was genome-based, this assumption is reasonable as the angle structure of elderly individuals with the angle-closure form is open-angle at birth, with the exception of congenital glaucoma, which presents with abnormal angle structure congenitally. The datasets for the summary statistics are described in detail in Table 1 and Figure 1.

### 2.3. Selection of the Genetic Instrumental Variables 

SNPs associated with iritis and uveitis at the genome-wide significance threshold (*p* < 5.0 ×$\;10^{-8}$) were used as the IVs. However, SNPs with *p* < 5.0 ×$\;10^{-4}$ were chosen as the IVs when there were no noteworthy thresholds, as the number of participants was limited in comparison to the rarity of the disease. SNPs were clumped using linkage disequilibrium (LD) with *R^2^* < 0.001 within 10,000 kb to ensure the independence of the IVs. The 1000 Genomes phase III dataset (East Asian) was used as the reference panel for computing LD for the clumping procedure. *F*-values were used to evaluate the strengths of the genetic IVs. The *F*-value was determined using the formula *F* = *R*^2^(*n* − 2)/(1 − *R*^2^), where *n* is the sample size and *R^2^* is the proportion of variances of exposure by the genetic variances [31]. *F*-values larger than 10 are regarded as no evidence of weak instrument bias [32]. 

### 2.4. Mendelian Randomisation

The MR analysis was performed under the following assumption: (1) IVs should have a significant relationship with the exposure; (2) IVs should not be linked to confounders of the exposure–outcome association; and (3) IVs should affect the outcome solely through exposure. We used the inverse-variance-weighted (IVW) method as our primary analysis method [32,33,34]. A fixed-effects model with three or fewer IVs was applied; a multiplicative random-effects model was used otherwise. Weighted median [35] and MR–Egger (with or without adjustment using the Simulation Extrapolation [SIMEX] method) regression [36,37] were considered the sensitivity analyses. The IVW approach has maximum efficiency when all genetic variations satisfy the three IV assumptions [38]. The estimate of IVW might be biased if one or more of the variants are invalid [35]. However, the weighted median technique generates consistent estimations of causality even when up to 50% of the IVs are invalid [35]. The MR–Egger method provides estimates of suitable causal effects even in the presence of pleiotropic effects by considering a non-zero intercept, which represents the average horizontal pleiotropic effects [36]. Bias can be corrected using MR–Egger regression with SIMEX when the assumption of no measurement error is violated (*I^2^* value < 90%) [37]. The Cochran’s Q statistic and Rücker’s Q′ statistic tests were used to assess the heterogeneity of the IVW and MR–Egger methods, respectively [33,39]. Pleiotropy in the genetic variant may be observed if the Cochran’s Q statistic and Rücker’s Q′ statistic tests have *p*-values less than 0.05. Directional horizontal pleiotropy was evaluated using the MR Polyhedral sum of residuals and outliers (MR-PRESSO) global test, and the presence of pleiotropic outliers was evaluated using the MR-PRESSO outlier test [40]. *p*-values of less than 0.05 for the MR-PRESSO global test and outlier test indicated possible pleiotropy in the genetic variations and the presence of pleiotropic outliers, respectively. All analyses were performed using the TwoSampleMR and simex packages in R version 3.6.3 (R Core Team, Vienna, Austria).

## 3. Results

### 3.1. Genetic Instrumental Variables

Two and eleven IVs at the significance level (*p* < 5.0 ×$\;10^{-8}$) were identified for iritis in the East Asian and multi-ethnic populations, respectively. Sixty and ten IVs at the significance level (*p* < 5.0 ×$\;10^{-4}$) were identified for uveitis in the East Asian and multi-ethnic populations, respectively (Table 2). The mean *F* statistics for iritis (41,588 for the East Asian population and 7704 for the multi-ethnic population) and uveitis (6433 for the East Asian population and 8396 for the multi-ethnic population) that were used for MR were greater than 10, demonstrating that there was a low chance of weak instrument bias (Table 2). Detailed information regarding the IVs utilized in this study is provided in Appendix A.

### 3.2. Heterogeneity and Horizontal Pleiotropy of Instrumental Variables

The IVW approach was employed as the IVs for iritis and uveitis were not heterogeneous in the Cochran’s Q test (all *p* > 0.05) (Table 2). In addition, the Rücker’s Q test from the MR–Egger test revealed no heterogeneity between IVs, and the MR–Egger regression intercepts showed that there was no horizontal pleiotropic effect before (all *p* > 0.05) and after SIMEX adjustment (all *p* > 0.05). These results indicated that there was no pleiotropic effect (Table 2). In addition, the MR-PRESSO global and outlier tests did not demonstrate horizontal pleiotropy (all *p* > 0.05; Table 2).

### 3.3. Mendelian Randomisation for the Casual Association between Iritis and Glaucoma 

The IVW method showed a significant causal association between iritis and glaucoma in the East Asian population (MR OR = 1.01, 95% confidence interval (CI): 1.00–1.01, *p* = 0.017, Figure 2). Similarly, the IVW method also showed a significant causal association between iritis and glaucoma in the multi-ethnic population (IVW MR OR = 1.04, 95% CI: 1.01–1.06, *p* = 0.001, weighted median MR OR = 1.05, 95% CI: 1.02–1.08, *p* = 0.003, MR–Egger MR OR = 1.05, 95% CI: 1.01–1.09, *p* = 0.028, and MR–Egger [SIMEX] MR OR = 1.05, 95% CI: 1.02–1.09, *p* = 0.020). Scatter plots show the genetic association with iritis against the genetic association with glaucoma for each SNP (Figure 3).

### 3.4. Mendelian Randomisation for the Casual Association between Uveitis and Glaucoma

The IVW method showed a non-significant causal association between uveitis and glaucoma in the East Asian population (MR OR = 0.9995, 95% CI: 0.9990–1.0000, *p* = 0.49, Figure 4). Since significant SNPs were not available (*p* < 5.0 ×$\;10^{-8}$), a less significant SNP (*p* < 5.0 ×$\;10^{-4}$) for uveitis in the East Asian population was chosen for analysis; therefore, the dependability of the analysis was low, as anticipated. In contrast, the IVW method showed a significant causal association between uveitis and glaucoma in the multi-ethnic population (IVW MR OR = 1.04, 95% CI: 1.02–1.06, *p* = 0.001, weighted median MR OR = 1.05, 95% CI: 1.02–1.08, *p* = 0.003, MR–Egger MR OR = 1.05, 95% CI: 1.01–1.08, *p* = 0.039, and MR–Egger [SIMEX] MR OR = 1.05, 95% CI: 1.01–1.08, *p* = 0.034). Scatter plots show the genetic association with uveitis against the genetic association with glaucoma for each SNP (Figure 5); a negative slope was observed for the East Asian population, whereas a significant positive slope was observed for the multi-ethnic population. 

## 4. Discussion

Our study demonstrates a causal association between iritis and glaucoma in an East Asian population and a multi-ethnic population. In addition, uveitis showed a causal association with glaucoma in the multi-ethnic population, although power constraints impeded the detection of this in the East Asian population. 

Research on the epidemiology of uveitis have shown results that are highly variable, primarily due to substantial methodological discrepancies. A recent systematic study on the frequency of uveitis showed prevalence numbers ranging from 9 to 730 occurrences per 100,000 and an overall incidence of 50.45 instances per 100,000, suggesting that geographical location is a significant driver of heterogeneity [41]. According to estimates, uveitis affects at least 2 million people worldwide and is a leading cause of blindness [42]. The inflammatory process that occurs within uveitis triggers the development of complications such as cataracts, macular edema, glaucoma, and retinal detachment [43]. According to one study, posterior synechiae (19.0%) was the most prevalent consequence of uveitis, followed by ocular hypertension (14.0%), macular edema (7.5%), and glaucoma (6.6%) [43]. Causal experimental studies on the induction of glaucoma by such severe uveitis have rarely been conducted. In this respect, this study is expected to be meaningful. Causal interference plays a major role in genetic epidemiology and clinical investigation and understanding the aetiology is implicit for identifying disease prevention and treatment opportunities. Existing epidemiological studies have limitations in the case of rare diseases; therefore, a prospective cohort study would be ideal. The MR is advantageous in this respect. Several MR studies have been conducted on ocular disease as a powerful approach to identify causal interference using human genetic data [44]. MR analysis has been used to investigate the impact of education levels, vitamin D, or medication on myopia; and lipid levels, refractive errors, or c-reactive protein levels on age-related macular degeneration [44]. In addition, lipid levels, central corneal thickness, type 2 diabetes, and refractive errors have been evaluated as risk factors for glaucoma in MR analyses [45,46,47]. From this perspective, the impact of iritis or uveitis exposure on the development of glaucoma can be investigated through a MR research approach.

The association between uveitis and glaucoma was first reported in 1813 as arthritic iritis followed glaucoma [2]. UG is one of the most common intraocular complications, and glaucoma after uveitis has an incidence of 11% after 5 years [10]. According to a Turkish epidemiologic study on 4604 eyes with glaucoma [48], UG accounted for 4.1% of all cases, and 92.4% of UG cases were OAG. In addition, glaucoma developed in 6.6% of patients with uveitis 1 year later [43]. While there have been numerous studies on the relationship between glaucoma and uveitis, the present study was performed as there was no data to support a causal relationship based on genetic determinants. Since uveitis is a rare disease and highly likely to be related to the *HLA* region, it is expected that fine-mapping analysis or next-generation sequencing will yield more desirable data. From this perspective, the exposure of SNPs associated with iritis and uveitis was confirmed to be involved in the occurrence of glaucoma, and a clear conclusion could be reached. SNPs, such as rs146683910, had a high positive beta value (beta = 14.87, Appendix A), which was related to the *HLA-B* locus associated with ankylosing spondylitis [49] (https://pheweb.org/UKB-TOPMed/pheno/715.2). An empirical study reported that *HLA-B* may be associated with OAG [50,51]. The causative analysis of uveitis and glaucoma also included immune-related genes in additional to *HLA*. The major histocompatability complex class I polypeptide-related sequence A *(MICA)* gene-related rs115681000 (beta = 1.732, located *MICA;LINC01149*) and rs146683910 (beta = 1.835, located *HLA-B;MICA-AS1*) also had high beta values (Appendix A). The *LINC01149* variation modifies the expression of *MICA*, making it easier for it to function as a key gene in the susceptibility to uveitis development. Nevertheless, despite the fact that the MR analysis used SNPs were primarily concentrated on chromosome 6, significant results were still seen despite the analysis’s usage of a relatively small number of SNPs due to the LD with *R*^2^ < 0.001 within 10,000 kb to ensure the independence of the IVs.

AAU is the most common type of uveitis, which is related to synechiae formation. Glaucoma, which is frequently associated with AS, psoriatic arthritis, and inflammatory bowel disease [52,53], had characteristics of autoimmune and inflammatory illnesses. In addition, a recent study using whole-exome sequencing identified rare variants and genes associated with IOP and glaucoma, including *BOD1L1*, *ACAD10*, and *HLA-B*, demonstrating the power of including and aggregating rare variants [54]. *HLA*-associated regions, such as *HLA-DRB1/DQA1*, *HLA-C,* and *HLA-DOA/HLA-DPA1,* were adopted while evaluating the SNPs utilized in our analysis (Appendix A). These loci showed significant association with uveitis, such as AAU, AS, and BD [5,13,49,52,53]. Moreover, significant SNPs were identified on comparing East Asian genetic data with the meta-analysis data, despite the ethnic differences. Nevertheless, our study demonstrated the value of using rare variants to enhance our understanding of the biological mechanisms regulating IOP and uncovered potential therapeutic targets for glaucoma [54], which we believe is consistent with our findings. Moreover, in our study, an association with glaucoma was found without intraocular pressure phenotype analysis, which further supports the findings. It is anticipated that future research will separate the description of disease into causal linkages rather than currently recognized categories.

The main strength of this study is the utilization of a relatively large Asian and European cohort dataset to reveal the causal association between iritis or uveitis and glaucoma. However, there are a few limitations of this study. First, we were unable to explain the presence of numerous confounding factors using summary statistics based on two-sample MR since we did not have access to individual-level data. Second, since iritis and uveitis are very infrequent disorders associated with immune-related (such as *HLA*) SNPs, they were not ideal study topics for MR analysis. However, the significant causality that emerged despite these limitations suggests that there is a very strong association. Third, there are test procedures to validate the MR hypotheses, but they do not provide complete validation. Therefore, as violation of MR assumptions can lead to invalid conclusions, the results should be interpreted with caution. 

## 5. Conclusions

We provide strong genetic evidence that supports a causal relationship between iritis and glaucoma in East Asian and multi-ethnic populations. Additionally, uveitis was found to have a causal effect on glaucoma in the multi-ethnic population, although there was limited causal evidence of this in the East Asian population owing to the lower power of IVs for that assumption. These causal correlations between iritis or uveitis and glaucoma imply that the current classifications of UG and OAG overlap, highlighting the need for additional research and caution in interpreting these complex interactions.

## Figures and Tables

**Figure 1 genes-14-00642-f001:**
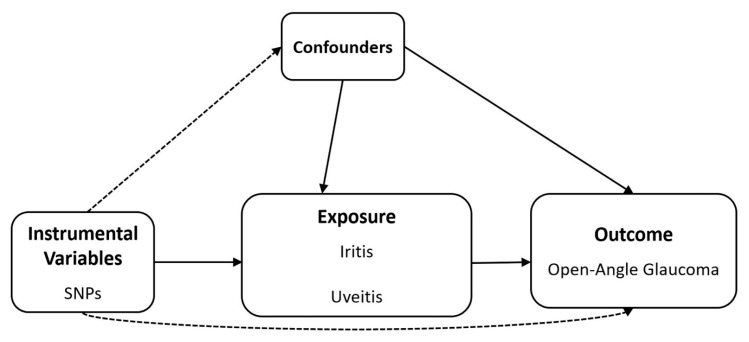
Schematic of the study design. SNP: single-nucleotide polymorphism.

**Figure 2 genes-14-00642-f002:**
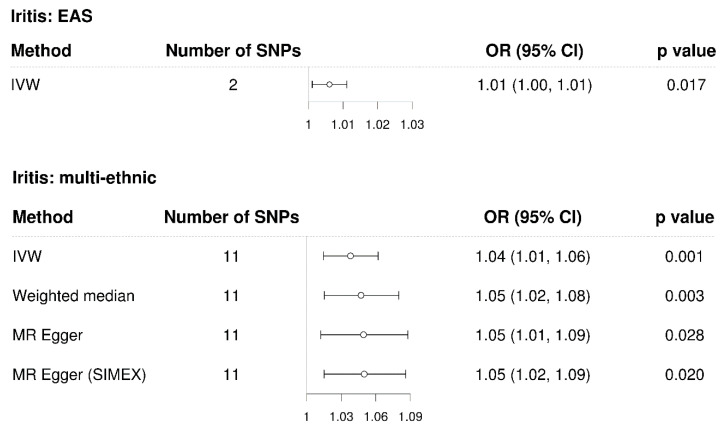
Forest plot of causal associations between iritis and glaucoma. EAS: East Asians; SNP: single-nucleotide polymorphism; OR: odds ratio; CI: confidence interval; IVW: inverse-variance weight; MR: Mendelian randomisation; SIMEX: simulation extrapolation.

**Figure 3 genes-14-00642-f003:**
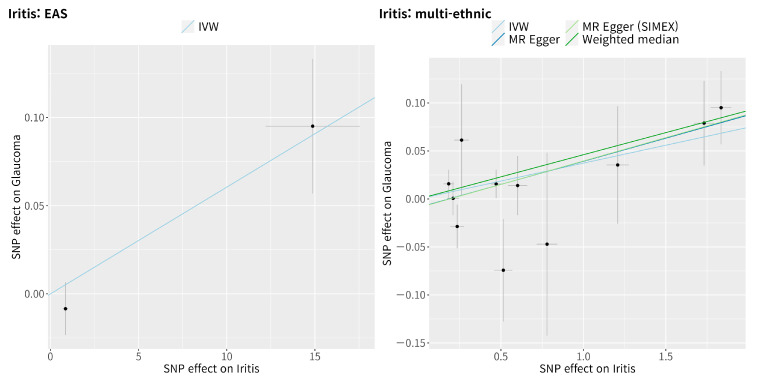
Scatter plots of Mendelian randomisation tests assessing the effect of the presence of iritis on glaucoma. The IVW, MR–Egger, MR–Egger (SIMEX), and weighted median estimate are indicated by the light blue, dark blue, light green, and dark green regression lines, respectively. SNP: single-nucleotide polymorphism; IVW: inverse-variance weight; MR: Mendelian randomisation; SIMEX: simulation extrapolation.

**Figure 4 genes-14-00642-f004:**
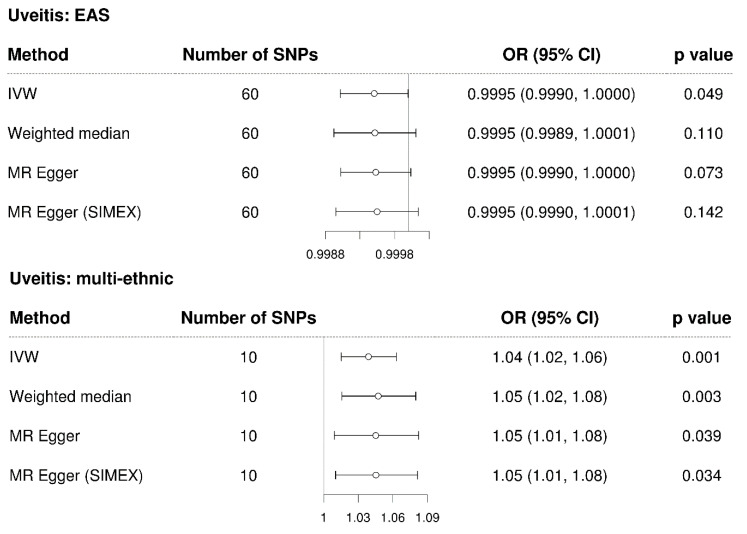
Forest plot of the causal associations between uveitis and glaucoma. EAS: East Asians; SNP: single-nucleotide polymorphism; OR: odds ratio; CI: confidence interval; IVW: inverse-variance weight; MR: Mendelian randomisation; SIMEX: simulation extrapolation.

**Figure 5 genes-14-00642-f005:**
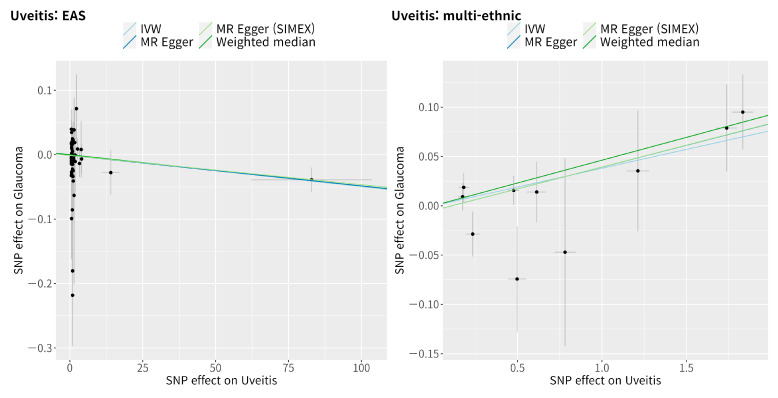
Scatter plots of the Mendelian randomisation tests assessing the effect of the presence of uveitis on glaucoma. The IVW, MR–Egger, MR–Egger (SIMEX), and weighted median estimate are indicated by the light blue, dark blue, light green, and dark green regression lines, respectively. SNP: single-nucleotide polymorphism; IVW: inverse-variance weight; MR: Mendelian randomisation; SIMEX: simulation extrapolation.

**Table 1 genes-14-00642-t001:** Summary statistics of the data sources.

Traits	Data Source	No. of Participants	Population	No. of Variants	Reference
Iritis	BBJ Project	175,653	East Asian	13,429,677	[29]
	BBJ Project + UKB	656,395	East Asian + European	25,844,196	[29]
Uveitis	BBJ Project	174,725	East Asian	13,429,518	[29]
	BBJ Project + UKB	655,467	East Asian + European	25,844,095	[29]
Glaucoma(Open-angle glaucoma)	GERA cohort + UKB	240,302 (12,315 cases + 227,987 control)	Multiethnic: 214,102 European 5103 African unspecified 3571 Other mixed ancestry 1847 African American or Afro-Caribbean 5189 Hispanic or Latin American 5370 East Asian 5120 South Asian	7,760,820	[20]

BBJ, BioBank Japan; GERA, Genetic Epidemiology Research in Adult Health and Aging; UKB, UK Biobank.

**Table 2 genes-14-00642-t002:** Heterogeneity and horizontal pleiotropy of instrumental variables.

Exposure				Heterogeneity		Horizontal Pleiotropy				
				Cochran’s Q Test from IVW	Rucker’s Q’ Test from MR-Egger	MR-PRESSO Global Test	MR–Egger		MR–Egger (SIMEX)	
	N	*F*	*I^2^* (%)	*p*-Value	*p*-Value	*p*-Value	Intercept, β (SE)	*p*-Value	Intercept, β (SE)	*p*-Value
Iritis	2	41588	96.33	0.359						
Iritis *	11	7704	99.11	0.587	0.558	0.659	−0.009 (0.011)	0.435	−0.009 (0.010)	0.389
Uveitis	60	6433	85.64	0.148	0.131	0.521	−0.001 (0.003)	0.688	−0.001 (0.003)	0.682
Uveitis *	10	8396	99.05	0.558	0.480	0.644	−0.005 (0.01)	0.644	−0.005 (0.010)	0.629

N, number of instrument variables; *F*, mean *F* statistic; IVW, inverse-variance weight; MR, Mendelian randomization; PRESSO, polyhedral sum of residuals and outliers; SIMEX, simulation extrapolation; β, beta coefficient; SE, standard error. *: indicates exposure using the SNPs selected from the meta-analysis from BioBank Japan Project and UK Biobank.

## Data Availability

The datasets for the genome-wide association study (GWAS) summary statistics can be found in the GWAS Catalog (https://www.ebi.ac.uk/gwas/summary-statistics, accessed on 19 July 2022).

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
