# Peer review of "Causal Association between Iritis or Uveitis and Glaucoma: A Two-Sample Mendelian Randomisation Study"

_genes, 2023, doi:10.3390/genes14030642_

Round 1
Reviewer 1 Report
I would like to thank the Authors of the Manuscript "Causal Association Between Iritis or Uveitis and Glaucoma: A Two-Sample Mendelian Randomisation Study" for the opportunity to provide commentary.
To my understanding, the Authors have revered data from large cohorts of East Asian ancestry, as well as combined East Asian and European ancestries, with the aim of testing the genetic determinants and association between iritis/uveitis and glaucoma using a two-sample MR approach.
Results show that several single nucleotide polymorphisms emerge as significantly related to an outcome of glaucoma in cohorts presenting iritis/uveitis, with the East Asian cohort and the multi-ethnic cohort providing different causal SNPs and, by extension, genes. A relevant observation is that many immune-related genes (specifically of the HLA family) appear as involved in this relationship and they are also characteristic of autoimmune and inflammatory illnesses.
To my understanding, this is an interesting study revealing pertinent genetic variants involved in the association and transition from iritis/uveitis to glaucoma, with a specific interest in East Asian populations that, in general, may not have been extensively studied before. The statistical methods used seem appropriate for the task at hand. My only suggestion to the Authors is to extend the Introduction and Discussion sections with more details regarding the variants/genes involved in iritis/uveitis and glaucoma, especially those that they found significantly associated, as this information is tucked in the Supplementary Material and no extra context is provided in the main text. English language may be checked as well, since several errors (due to distraction, I am sure) may be found scattered throughout the text.
Author Response
Response: We would like to thank the Reviewer for their careful review of our manuscript and their valuable suggestions. In accordance with your suggestion, we have added further details regarding the variants/genes involved in the Introduction and Discussion sections. In addition, we have made use of an English editing service to re-edit our manuscript. We have attached the certificate for English editing for your perusal accordingly.
Lines 53–56 “Recent investigations on iritis and uveitis have indicated that uveitis is characterised by a strong relationship with the human leucocyte antigens (HLA) gene. Moreover, acute anterior uveitis (AAU), Behcet’s disease (BD), and Vogt-Koyanagi-Harada were found to be highly correlated with HLA-B27, HLA-B51, and HLA-DR4/DQA1, respectively [5,13].”
Line 121 “The 1000 genomes phase III dataset (East Asian) was used as the reference panel for computing LD for the clumping procedure.”
Lines 256–259 “AAU is the most common type of uveitis, which is related to synechiae formation. Glaucoma, which is frequently associated with AS, psoriatic arthritis, and inflammatory bowel disease [49,50], had characteristics of autoimmune and inflammatory illnesses.”
Lines 262–267 “HLA-associated regions, such as HLA-DRB1/DQA1, HLA-C, and HLA-DOA/HLA-DPA1, were adopted while evaluating the SNPs utilized in our analysis (Table S1). These loci had shown significant association with uveitis, such as AAU, AS, and BD [5,13,46,49,50]. Moreover, significant SNPs were identified on comparing East Asian genetic data with the meta-analysis data, despite the ethnic differences.”
Reviewer 2 Report
I read the paper entitled ”Causal Association Between Iritis or Uveitis and Glaucoma: A Two-Sample Mendelian Randomisation Study” very carefully and concluded that the paper is acceptable in the present form for publication in your journal. The topic of the article is interesting. The findings of this study contribute to causal association between iritis or uveitis and glaucoma and to further investigations of this relationships. The authors concluded that interpretation should be with care.
Author Response
Response: We would like to thank the reviewer for their careful review of our manuscript and their encouraging comment. Please note that we have made the following change in the manuscript.
Lines 291–292: “the current classifications of UG and OAG overlap, highlighting the need for additional research and caution in interpreting these complex interactions.”